# Cold Atmospheric Plasma Jet Irradiation Decreases the Survival and the Expression of Oncogenic miRNAs of Oral Carcinoma Cells

**DOI:** 10.3390/ijms242316662

**Published:** 2023-11-23

**Authors:** Yun-Chien Cheng, Kuo-Wei Chang, Jian-Hua Pan, Chao-Yu Chen, Chung-Hsien Chou, Hsi-Feng Tu, Wan-Chun Li, Shu-Chun Lin

**Affiliations:** 1Department of Mechanical Engineering, College of Engineering, National Yang Ming Chiao Tung University, Hsinchu 300093, Taiwan; yccheng@nycu.edu.tw (Y.-C.C.); tacocy@hotmail.com (C.-Y.C.); 2Institute of Oral Biology, College of Dentistry, National Yang Ming Chiao Tung University, Taipei 112304, Taiwan; ckcw@nycu.edu.tw (K.-W.C.); g00750@nycu.edu.tw (J.-H.P.); michaelchou0806@gmail.com (C.-H.C.); hftu@nycu.edu.tw (H.-F.T.); wcli@nycu.edu.tw (W.-C.L.); 3Department of Dentistry, College of Dentistry, National Yang Ming Chiao Tung University, Taipei 112304, Taiwan; 4Department of Stomatology, Taipei Veterans General Hospital, Taipei 112304, Taiwan

**Keywords:** AKT, cold atmospheric plasma, ERK, miRNA, oral carcinoma

## Abstract

Despite recent advancements, therapies against advanced oral squamous cell carcinoma (OSCC) remain ineffective, resulting in unsatisfactory therapeutic outcomes. Cold atmospheric plasma (CAP) offers a promising approach in the treatment of malignant neoplasms. Although the effects of CAP in abrogating OSCC have been explored, the exact mechanisms driving CAP-induced cancer cell death and the changes in microRNA (miRNA) expression are not fully understood. We fabricated and calibrated an argon-CAP device to explore the effects of CAP irradiation on the growth and expression of oncogenic miRNAs in OSCC. The analysis revealed that, in OSCC cell lines following CAP irradiation, there was a significant reduction in viability; a downregulation of miR-21, miR-31, miR-134, miR-146a, and miR-211 expression; and an inactivation of the v-akt murine thymoma viral oncogene homolog (AKT) and extracellular signal-regulated kinase (ERK) signals. Pretreatment with blockers of apoptosis, autophagy, and ferroptosis synergistically reduced CAP-induced cell death, indicating a combined induction of variable death pathways via CAP. Combined treatments using death inhibitors and miRNA mimics, alongside the activation of AKT and ERK following the exogenous expression, counteracted the cell mortality associated with CAP. The CAP-induced downregulation of miR-21, miR-31, miR-187, and miR-211 expression was rescued through survival signaling. Additionally, CAP irradiation notably inhibited the growth of SAS OSCC cell xenografts on nude mice. The reduced expression of oncogenic miRNAs in vivo aligned with in vitro findings. In conclusion, our study provides new lines of evidence demonstrating that CAP irradiation diminishes OSCC cell viability by abrogating survival signals and oncogenic miRNA expression.

## 1. Introduction

Head and neck squamous cell carcinomas (HNSCCs), which include oral SCCs (OSCCs), rank as the sixth most prevalent malignancy globally [1,2]. Multiple molecular aberrations have been shown to modulate the neoplastic progression of HNSCC/OSCC [1,2,3]. The phosphoinositide 3-kinases (PI3K)/v-akt murine thymoma viral oncogene homolog (AKT)/mammalian target of the rapamycin (mTOR) signaling complex orchestrates diverse regulatory functions in cell survival, disease pathogenesis, and tumorigenic processes [4]. Remarkably, this crucial axis is disrupted in 90% of HNSCC cases [1]. Extracellular signal-regulated kinase 1/2 (ERK1/2) also plays central roles in the anti-apoptotic pathways, proliferation, and drug resistance of HNSCC [4,5]. Additionally, microRNAs (miRNAs) have a function in the neoplastic process, including in the pathogenesis of HNSCC/OSCC. The dysregulation of oncogenic miRNAs, such as miR-21, miR-31, miR-134, miR-146a, miR-187, and miR-211, and the suppressor miR-125b, together with the disruption of their targets in OSCC, has been identified in previous studies [6,7,8,9,10,11,12,13]. Notably, some of these miRNAs are also ROS modulators or signal effectors [11,12].

Plasma is an ionized gas and the fourth state of matter. Cold atmospheric plasma (CAP) is produced by inert gasses or air under atmospheric pressure at room temperature. CAP comprises atoms, electrons, ions, molecules, and reactive oxygen and nitrogen species (RONS) [14]. At elevated doses, RONS intensify oxidative stress, induce DNA damage, increase cytotoxicity, and promote tissue pathogenesis [15]. CAP’s biological effects have found medical applications in enhancing wound healing, disinfection, and skin disease control and in aiding cancer therapies [16,17,18]. The anti-tumor efficacies of CAP in enriching cell cycle arrest, inducing apoptosis, enhancing tumor chemosensitivity, eliminating cancer stemness, and modulating the tumor microenvironment have been shown in multiple malignancies [14,19,20,21]. Distinct CAP responses between neoplastic cells and their normal counterparts highlight the potential utility of CAP in cancer treatment [22,23,24,25]. RONS, such as hydrogen peroxide (H_2_O_2_), may induce apoptosis in some cancer cell lines. Other RONS that can possibly affect cancer cells include O, H_2_O_2_, •O_2_^−^, •OH, NO, ONOO^−^, N_2_, N^+^, Argon (Ar), Ar^+^, and Ar^2+^. Additionally, NO can both improve and inhibit the apoptosis of cells, but the mechanism is still not yet clear [26,27,28,29,30]. CAP reduces the viability of cancer cells by impairing multiple RONS-dependent or -independent signals [17,25,31,32,33,34,35,36,37,38]. Apart from apoptosis, a diverse profile of programmed death, including ferroptosis, autophagy, pyroptosis, and necroptosis, could also be evoked by CAP in different types of cells [25,38,39,40,41,42,43]. Interestingly, CAP-associated lysosomal NO induces the switch from autophagy to ferroptosis in cancer cells [44]. Recent advances in detecting the oxidative stress, flipped annexin V, autophagosome, or intracellular ion in live cells have enhanced insights into the CAP effects or the interplay amongst death pathways [29,43,44,45,46,47].

The promising in vitro and in vivo effects of CAP in abrogating HNSCC or OSCC have been explored [18,19,22,32,33,34,35,43,48,49]. As it is more superficially located, oral cavity tissue is highly vulnerable to non-invasive CAP assessments [50]. While initial human trials on advanced OSCC have shown tumor flattening or ulcer rim contraction in areas treated with a CAP jet [51], a broader CAP trial is needed to establish its clinical efficacy. CAP induces AKT degradation by activating the ubiquitination–proteasome system, which leads to AKT inactivation and apoptosis in HNSCC [34]. CAP seems to activate pro-apoptotic p38 MAPK and JNK in different cell types, including HNSCC cells, albeit with no effects on ERK [31,33,36]. CAP also deactivates NF-κB and reduces the associated phenotypes in some OSCC cells [32]. CAP irradiation modifies miRNA profiles in myeloid cancer [17]; its role in modulating miRNA expression in OSCC is yet to be fully understood.

Despite the advances in mechanistic understanding and therapeutic approaches, the survival rate of patients with HNSCC/OSCC has not drastically improved in recent decades [2,6,11]. Further effective treatments are required for improving patient survival. Since the combination of CAP and chemotherapeutic reagents results in remarkable HNSCC inhibition [32,52], together with the differentiation in sensitivity to CAP across normal and cancer cells following the implementation of CAP [22,31,35], CAP irradiation could be considered an adjuvant therapeutic strategy against HNSCC/OSCC. This study investigated the altered expression of vital miRNAs, particularly the recognized oncogenic miRNAs in OSCC and cell lines after CAP irradiation. The effects of CAP on signal elements in modulating tumor survival were also investigated. We identified for the first time the downregulation of a panel of oncogenic miRNAs secondary to the inactivation of AKT/ERK signals mediated by Ar-CAP irradiation. The preclinical tests also highlight the anti-tumor potential of CAP irradiation. Our research provides new molecular perspectives on the CAP mechanisms that inhibit OSCC.

## 2. Results

### 2.1. CAP Structure, the CAP-Induced RONS, and Cell Death

We fabricated Ar-CAP jet in our laboratory (Figure 1A). The structure of our self-made Ar-CAP jet was a cylindrical dielectric sandwiched between two electrodes (Figure 1A). The cylindrical dielectric was a quartz tube (dielectric constant: 3.9). The grounding electrode was a stainless-steel rod connected to a slot-shaped electrode, and it was placed in the quartz tube. Aluminum tape was attached to the outside of the tube as the high-voltage electrode. Sine waves were used to ignite the plasma with a frequency of 20 kHz. The peak voltage was 2.77 kV. The Ar flow rate was 3 slm. The RONS generated by plasma in the gas phase were measured using an optical emission spectrometer (OES, PI Acton SP 2500, Princeton Instruments, Trenton, NJ, USA). The spectrum peaks show that our CAP generated the OH radical, nitrite, and excited Ar in phosphate-buffered solution (PBS) (Figure 1B). The OES integration time was 100 ms and the focal length was 3.5 cm in our measurement system. According to a previous study, oxidative stress is not the only factor that causes cell responses. Other species, such as radicals or RNS, also have an impact on cells [53,54,55,56]. However, it is difficult to measure all these species, so we only measured H_2_O_2_ and nitrite as reference values in this study.

The H_2_O_2_ concentration increased with the 10–60 s CAP irradiation, and the four H_2_O_2_ concentration curves, which were measured every 10 min, showed no significant difference. This result indicates ROS reproducibility in our CAP-treated solution (Figure 1C). An increase in nitrite following the 10–30 s CAP irradiation was also consistently noted across all time durations (Figure 1D). The nitrite production following 60 s CAP irradiation at a 40–46 min time duration was slightly lower than that in the shorter time durations. The production of nitrate was not detectable (not shown). Normal oral keratinocytes (NOKs) and the OSCC cell lines SAS and FaDu were treated with CAP for 5–60 s. The viability assays showed a dose-dependent inhibition of the growth of NOK, FaDu, and SAS mediated by CAP irradiation (Figure 1E). SAS exhibited the highest CAP sensitivity, while NOK exhibited the lowest CAP sensitivity. FaDu exhibited modest survival responses following CAP irradiation, and ROS induction was illustrated using fluorescence assays (Figure 1F).

### 2.2. CAP Induces Apoptosis, Autophagy, and Ferroptosis of OSCC Cell Lines

CAP-induced cell death (10 s irradiation) was partly mitigated in SAS and FaDu cells by individually pretreating them with 10 µM Z-VAD-FMK (Z-VAD), 3-Methyladenine (3-MA), and ferropstatin-1 (Fer-1) (Figure 2A Lt; Figure 2B Lt, respectively). Pretreatment with 20 µM inhibitors mitigated the CAP-associated cell death in the cells (Figure 2A Rt and Figure 2B Rt). The combined treatment of two inhibitors resulted in a more pronounced reversion of cell death than solitary treatment. The combined treatment with 10 µM of all inhibitors restored cell survival to 83–92%, while the combined treatment with 20 µM of all inhibitors restored cell viability to around 93% (Figure 2A,B). The effects of Z-VAD and Fer-1 were also identified in NOK (Figure 2C). The rescue of cell viability in the 10 s CAP treated SAS cells by inhibitors appeared to be dose-dependent (Figure 2D). Moreover, following the 5–20 s CAP irradiation, the viability of SAS cells was partially rescued when each inhibitor was used alone. The concomitant treatment of the three inhibitors resulted in a more prominent rescue of cell survival (Figure 2E). Although the concordant induction of apoptosis, autophagy, and ferroptosis may underlie the CAP-induced death of OSCC cells, the rescue effects of the ferroptosis inhibitor were more potent than those of the other two inhibitors. Additionally, 3-MA tended to have weaker rescue effects than the other inhibitors, which might suggest a mild induction of autophagy by CAP in the tested cells. The induction of variable types of cell death following the 10 s CAP irradiation was confirmed via the fluorescence detection of flipped phosphatidylserine on the cell membrane, the appearance of scattered autophagosomes, and the pronounced accumulation of cytosolic ferrous iron representing the enrichment of the ferroptotic microenvironment in FaDu cells compared to controls (Figure 2F).

### 2.3. CAP Downregulates the Expression of Oncogenic miRNAs in OSCC Cells

Following 10 s CAP irradiation over 24 h, we observed a downregulation in the expressions of miR-21, miR-134, miR-146a, and miR-211, while miR-125b expression was upregulated in NOK, SAS, and FaDu cells (Figure 3A). Although miR-31 was downregulated in the OSCC cell line following such treatment, it was upregulated in NOK (Figure 3A). Treatment with 5 s CAP also resulted in the dysregulation of miRNA with lesser changes in expression levels (Figure 3A). The effects of the 10 s CAP irradiation at 24 h also lasted to 48 h (Figure 3B).

### 2.4. NaNO_2_ Treatment Simulates CAP-Induced miRNA Modulation

As the 5–10 s CAP irradiation yielded 12.5 µM H_2_O_2_ and 2 µM NO_2_^−^, respectively (Figure 1C,D), the cells were first treated with 12.5 µM H_2_O_2_ for 1 h to simulate the CAP-induced states. At 24 h, the assay showed no remarkable change in miRNA expression (Appendix A). However, the prolonged treatment of SAS cells with 12.5 µM H_2_O_2_ for 24 h resulted in decreased miR-21, miR-125b, miR-134, and miR-211 expressions, along with increased miR-146a expression and no change in miR-31 expression in SAS cells (Appendix A). The treatment with 2 µM NaNO_2_ for 1 h caused decreased miR-21 and miR-211 expressions in the cells (Figure 3C). Cotreatment with H_2_O_2_ was unable to augment the influences drastically. Following the NaNO_2_ treatment, miR-31 expression increased in NOK, while miR-31 expression decreased in SAS and FaDu cells.

### 2.5. The Downregulation of Oncogenic miRNAs Underlies the CAP-Induced Decrease in Cell Survival

To investigate whether the alteration in miR-31 expression influences the varied CAP responses across cells, we either downregulated or upregulated miR-31 expression in NOK using an inhibitor or a mimic, respectively (Figure 4A). As the higher miR-31 expression levels were associated with the better survival of NOK (Figure 4B), the upregulation of miR-31 following CAP irradiation in NOK may underlie its relatively lower lethality in response to CAP irradiation. With the pretreatment of miRNA mimics to upregulate miR-21, miR-31, miR-146a, and miR-211, the survival of OSCC cells was higher than that of control cells after the 10 s CAP treatment (Figure 4C). Since the treatment with the miR-31 mimic and miR-211 mimic rendered a higher cell survival than the other mimics, we tested the efficacy of cotreatment. The cotreatment of the miR-31 mimic and the miR-211 mimic restored the survival of OSCC cells after 10 s and 15 s CAP irradiations up to 98% and 76%, respectively (Figure 4D). The CAP-modulated miRNA expression underlies the changes in cell viability.

### 2.6. CAP Inactivates AKT and ERK to Reduce Cell Survival and Impede miRNA Expression

SAS cells display a modest level of endogenous ERK activation and lack AKT expression. To test the effects of CAP on signal activation, SAS cells were transfected with AKT or ERK plasmids for exogenous overexpression. The transient overexpression also accompanied remarkable AKT and ERK activation (Figure 5A, Appendix A). Notably, CAP irradiation did not affect the abundance of AKT or ERK proteins; regardless of endogenous or exogenous overexpression, it reduced endogenous ERK activity and exogenous AKT and ERK activity 2 h after treatment. The treatment with 10 s CAP caused more conspicuous ERK inactivation than the treatment with 5 s CAP (Appendix A). The increase in cell survival associated with AKT or ERK activation was attenuated by 10 s CAP (Figure 5B), implying the involvement of these signals in CAP-induced cell death. The activation of AKT and ERK also drove alterations in miRNA expression (Figure 5C–E). The upregulation of miR-21, miR-31, miR-187, and miR-211 associated with AKT or ERK activation was significantly attenuated by 10 s CAP (Figure 5C). miR-134 expression was upregulated by AKT activation but downregulated by ERK activation (Figure 5D). Although miR-146a expression was drastically upregulated by AKT or ERK activation, the downregulation of 10 s CAP on miR-146a expression was pronounced, and this was not associated with the activation of these signals (Figure 5E).

### 2.7. CAP Decreases the Xenograft Tumor Growth of the SAS Cell Line

Xenografts of SAS cells were divided into five groups receiving various treatments and assays (illustrated in Figure 6A). Single-shot CAP irradiation for 10 min or 14 min decreased the tumor volume to 67% relative to that of the controls. Two shots of CAP irradiation for 10 min/10 min or 14 min/14 min decreased the tumor volume to 42% of that of the control tumors. The CAP irradiation generally decreased tumor volume to 54% compared to that of the argon-treated controls (Figure 6B). Neither the body weight of mice (Figure 6C) nor the serological data of liver and renal function (Figure 6D) changed at the endpoint (7.0 weeks).

A histopathological examination of the control tumor revealed aggregations of Ki-67-stained cells, focal areas of necrosis, and extensive inflammatory cell infiltration. These tumors were enclosed by a slender fibrovascular connective tissue layer interspersed with epithelioid cells (Figure 6E). The histopathological features of the treated tumors were largely the same as those of the control tumors despite a lower Ki-67-labeled cellular fraction, a thicker fibrovascular epithelial cell layer, and more extensive necrosis. The histopathological evaluation of tissue sections from two autopsy mice receiving the highest dose of CAP irradiation revealed no remarkable tissue changes in the tongue, lung, liver, or kidney (Appendix A). CAP irradiations caused no histopathologically discernable skin injury in the mice (Appendix A).

Compared to the controls, the tumors that underwent a single CAP irradiation displayed varied extents of miRNA downregulation (Figure 6F, Appendix A). The decrease in miR-21 and miR-211 was particularly eminent. miR-125b expression was also downregulated in the CAP-treated tumors. A Western blot analysis indicated abundant endogenous AKT and ERK expressions, drastic AKT activation, and the complete absence of ERK activity in the tumors (Figure 6G, Lt). A site-by-site analysis demonstrated a lack of ERK activity in the xenografts compared to cells in culture (Figure 6G, Rt). The tumors receiving one-shot CAP irradiation did not exhibit a change in AKT activation, whereas the tumors receiving two-shot CAP irradiation exhibited decreased AKT activation (Figure 6G, Lt).

## 3. Discussion

Several studies have documented the efficacy of CAP in mitigating the effects of OSCC and HNSCC [18,19,22,32,33,34,35,43,48,49]. Our previous studies demonstrated equal effects across direct He-CAP irradiation exposure and a He-CAP-treated medium in abrogating the growth of cancer cells [20]. Additionally, the He-CAP-treated medium synergized with anti-cancer drugs to induce cancer death. Although differentiation in sensitivity to CAP across normal and cancer cells has been found [22,31,35], a cross-comparison of CAP effects in different types of cells or culture conditions may raise concerns regarding bias. In this study, we developed an argon-CAP jet and characterized its efficacy and stability. By evaluating identical cell types (squamous cells) grown in a consistent medium, our study confirms that normal oral keratinocytes are less sensitive to CAP toxicity than OSCC cell lines. As a factor contributing to the immortalization and neoplastic transformation of oral keratinocytes [9,11], a mild increase in miR-31 expression following CAP irradiation to sustain the growth capability of NOK may be a potential explanation for such a discrepancy in CAP sensitivity. Therefore, CAP could be a promising strategy to intercept OSCC, which causes less toxicity to the surrounding normal oral epithelium.

The influence of CAP in inducing apoptotic cell death is well documented in various malignancies, including HNSCC [33,52]. Recent studies have revealed that CAP also causes the autophagic and ferroptotic death of cancer cells [41,43,48]. The experiments in this study show that the rescue of viability by a ferroptosis blocker was much more pronounced than that by inhibitors of apoptosis or autophagy. These findings, together with the findings of a drastic increase in intracellular ferrous ion, substantiate the effects of CAP-associated RNS in ferroptosis induction [24], while the orchestration of apoptosis and autophagy death also exists. Although crosstalk occurs amongst various types of programmed death [57,58], since combined treatment with 20 µM of ferroptosis/apoptosis/autophagy inhibitors nearly abolished the cell mortality induced by the 10 s CAP irradiation, the involvement of additional types of CAP-induced cell death in OSCC could be limited [38]. The reinforcement of the ferroptosis death induced by CAP using drugs may facilitate OSCC interception [41].

CAP’s impact on miRNA expression has been previously noted exclusively in leukemia cells [17]. For the first time, we identified the downregulation of various oncogenic miRNAs and the upregulation of miR-125b in OSCC cells, influenced by CAP. In accordance with previous functional studies, such miRNA disruption may underlie the oncogenic suppression [6,7,8,9,10,11,12,13]. As the treatments with miRNA mimics partially restored the cell viability repressed by CAP, our study provides clues substantiating that the disruption of miRNA expression could be the mechanism of the CAP-induced lethality of OSCC cells. Our preliminary findings suggest an association between NO_2_^−^ and the disruption of miRNA expression. The mechanisms underlying the CAP-induced products and the aberrance in miRNA regulation require further investigation. Since this study only analyzed selected miRNA targets that were previously studied [6,7,8,9,10,11,12,13], upon gaining an insight of a comprehensive landscape of miRNA expression affected by CAP, a strategy combining CAP and miRNA targeting could be validated to eliminate cancer cells.

Research indicates that CAP diminishes AKT levels and stimulates p38 and JNK in HNSCC cells [33,34]. However, the effects of CAP in modulating ERK proteins or activity are yet to be addressed. The endogenous AKT and ERK expressions were minimal or modest in SAS cells under our culture conditions, while the transfection of plasmids induced drastic exogenous protein expression and consequential activation. Since the 10 s CAP irradiation completely abolished pAKT and pERK, along with the fact that AKT and ERK activation paralleled the cell survival and the expressions of miR-21, miR-31, miR-187, and miR-211, the CAP-AKT/ERK-miRNA-survival cascade could be the anti-OSCC mechanism of CAP. Although CAP repressed miR-134, the influences of AKT and ERK on miR-134 expression were different. Both AKT and ERK upregulated miR-146a, but CAP-mediated miR-146a downregulation was irrelevant to these signals. This study demonstrates the conspicuous effects of CAP in suppressing the activity of AKT/ERK, which are critical factors of neoplastic survival. The phosphatase and tensin homolog (pTEN) is a potent inhibitor of PI3K/AKT/mTOR-mediated tumor suppression [39]. Dual-specificity phosphatases (DUSPs), such as MKP1 (DUSP1), dephosphorylate the serine/threonine residues of ERK to mediate tumor suppression [59,60]. The regulatory action of CAP or CAP products on pTEN and DUSPs or their effects in disturbing the chemical bonds of signal molecules need to be addressed to discover the inactivation mechanism. The effects of CAP on the abundance of signaling factors in OSCC cells require wider stratification [33,34]. As SAS displayed rather low endogenous ERK and AKT activities under our experiment conditions, together with its insidiously high transfection efficiency and sensitivity to CAP [43], we established a cell model displaying the abundance activation of survival signals via exogenous expression to test the CAP effects [6,11]. This platform could be further extended to fully elucidate the effects of CAP on other signaling pathways or oncogenic phenotypes [43].

We explored the in vivo implications of CAP using a nude mouse model. For consistent efficacy, we employed the same jet appliance for irradiation as used in our in vitro experiments. Although the CAP regimens in our study are different from those in other studies [33,34], the inhibitory effects of single or double CAP irradiation on SAS xenografts were evident, and no local or systemic toxicities were detected. Despite the fact that Ki-67 immunoreactivity seemed unchanged in the surviving cells of shrunk tumors, the changes in miRNA expression profiles generally occurred in concert with those that occurred in the in vitro experiments. Both total AKT and ERK proteins seemed to be more abundant in the tumors than in the original cultivated cells. AKT activation appeared to be synchronized with the total AKT amount, but the modest ERK activation in the original cells was completely silenced in the tumors. The mild inactivation of AKT in the tumors that received two-shot CAP irradiation may be due to the limited irradiation dose or the limited penetration distance of CAP and CAP-induced compounds. Otherwise, the decrease in AKT activation could be secondary to the tumor shrinkage. Since CAP therapy would be more effective in the superficial compartment of tumors, a precise preclinical regimen of CAP irradiation should be developed to achieve the effective elimination of early-stage OSCC. Further studies are required to better demonstrate the influence of CAP on the tumor microenvironment, especially on the immune microenvironment in a syngeneic murine OSCC model [34].

While therapeutic trials involving CAP in human OSCC are underway [50,51], a deeper mechanistic understanding of CAP is essential to bolster theoretical foundations and refine practical guidelines. Our research indicates that CAP acts as a potent suppressor of survival signals and oncogenic miRNAs in OSCC, primarily by triggering multiple forms of programmed cell death. In our preclinical tests, CAP irradiation also effectively inhibited the xenografic growth of OSCC. The anti-tumor potential plus the safety of this therapeutic approach may enable it to be a potent adjuvant to OSCC therapy, particularly for tumors exhibiting less depth and harboring high AKT/ERK activation or high oncogenic miRNA expression.

## 4. Materials and Methods

### 4.1. Plasma Jet Fabrication and RONS Measurement

A CAP jet was fabricated in our laboratory (Figure 1A). The parameters of the device are described in the Figure 1A. After optimizing the CAP jet established in our laboratory, we analyzed the RONS emission. The RONS in CAP was measured using an OES. The 309 nm, 300–400 nm, and 700–800 nm peaks of the OES spectrum measured from CAP represent the OH radical, RNS, and argon emission, respectively. The preliminary tests of the efficacy and stability of this system were performed in 1xPBS [30]. To test the production of soluble ROS/RNS and the stability of the appliance, we analyzed the H_2_O_2_ and nitrate/nitrite concentrations in PBS after 10 s, 20 s, 30 s, and 60 s CAP irradiation was performed; the measurements were performed every 10 min, and CAP was switched on for 46 min during the measurement. H_2_O_2_ and nitrate/nitrite were measured using an Amplex^®^ Red Hydrogen Peroxide/Peroxidase Assay Kit (Thermo Fisher Scientific, Waltham, MA, USA) and a Nitrate/Nitrite Colorimetric Assay Kit (Cayman, Ann Arbor, MI, USA), respectively.

### 4.2. Cell Culture, Reagents, and Plasmid Transfection

The SAS cell line (Tongue SCC, RCB1974 in Japanese Collection of Research Bioresources Cell Bank, Tokyo, Japan) was routinely cultivated in DMEM (Biochrom AG, Burlington, MA, USA) supplemented with 1% L-glutamine (Biological Industries, Tel Aviv, Israel). The FaDu cell line (Hypopharyngeal SCC, HTB-43^TM^ in the American Type Culture Collection, Manassas, VA, USA) was grown in Minimum Essential Medium (Gibco, Carlsbad, CA, USA) with 1% sodium pyruvate (Gibco). The media were supplemented with 10% fetal bovine serum (FBS, Biological Industries). These cell lines are frequently used in HNSCC/OSCC studies, including CAP research [5,6,11,43]. NOK [9], a human telomerase reverse-transcriptase gene immortalized normal oral keratinocyte that we established previously, was routinely cultured in keratinocyte serum-free medium (KSFM, Invitrogen, Carlsbad, CA, USA). All culture media also contained 1% 3-in-1 antibiotics (100 unit/mL penicillin, 0.1 mg/mL streptomycin, 250 ng/mL amphotericin, Biological Industries). As these cells grew and attached stably in a mixed DMEM/KSFM (1:1) culture medium within five days in pilot tests; they were all transferred to this medium to grow in 96-well cultivation plates (Thermo Fisher Scientific) 20–24 h before the CAP experiment to ensure that there were no discrepancies in the cultivation conditions for the CAP experiments (Figure 1A). One hour after CAP irradiation, the culture medium was replaced. The cell viability was measured 24 h later using a trypan blue exclusion assay. Treatment with argon for 60 s without electricity was the control.

The RONS simulators H_2_O_2_ and NaNO_2_, the pan-caspase inhibitor Z-VAD that alleviates apoptosis, the autophagy/PI3K inhibitor 3-MA, the ferroptosis inhibitor Fer-1, and dimethyl sulfoxide (DMSO) were purchased from Sigma-Aldrich (St Louise, MO, USA). Hank’s balanced salt solution (HBSS) was purchased from Thermo Fisher Scientific. The plasmids ERK2 (MAPK1; CAT#: 39223, Addgene, Watertown, MA, USA) and pUSE-AKT (a gift from Professor Yang, C.C.) were used for signal activation by means of transient overexpression as related to their vector-alone (VA) control [12]. TransFectin^TM^ Reagent (BioRad, Hercules, CA, USA) was used for transfection. Unless specified, all other materials were also purchased from Sigma-Aldrich.

### 4.3. Detection of Cellular ROS, Apoptosis, Autophagy, and Ferroptosis

The cells irradiated with CAP were washed, and then they were incubated with kits and reagents for the fluorescent detection of ROS using DCFDA, flipped phosphatidylserine using labeled Annexin V, autophagosomes, and intracellular ferrous ion, according to the protocols provided by the manufacturers (Appendix A) [29,43,45,46,47]. The reaction conditions are also detailed in Appendix A. After the reaction, the cells were washed, and fluorescence images were captured using an inverted microscope (Eclipse, TS100, Nikon, Tokyo, Japan).

### 4.4. Treatment with Cell Death Inhibitors

Z-VAD, 3-MA, and Fer-1 were dissolved in DMSO and optimized to 10 mM stock solution for use. Cells were pretreated with 10 µM or 20 µM of these inhibitors for 1 h prior to receiving 5–20 s CAP irradiation [60,61,62,63].

### 4.5. Treatment with miR-31 Inhibitor and miRNA Mimics

The following are listed in Appendix A: the inhibitor of miR-31; the mimics of miR-21, miR-31, miR-146a, and miR-211; and their controls. They were optimized to 60 nM for use [6,11]. Cells were transfected with the miR-31 inhibitor or miRNA mimics using 0.75 µL/mL. After transfection, cells were harvested 24 h later and re-seeded for 20 h to receive 10 s or 15 s CAP irradiation.

### 4.6. Animal Studies

For the induction of xenografts, 10^6^ SAS cells were mixed with 40% Matrigel^®^ Matrix (Corning, Bedford, MA, USA) and intradermally injected into the flanks of nude mice (National Laboratory Animal Center, Taipei, Taiwan). The tumor volume was calculated in accordance with the following formula: volume = 0.5 ab^2^, where a is the longest diameter and b is the shortest diameter [11]. To acquire an adequate amount of tissue samples for a comprehensive analysis, the tumors were grown to diameters exceeding >1 cm, which was achieved by the 6th week. The body weights of the mice were measured weekly. CAP jet irradiation was performed on the surface of the tumor center or non-tumor-bearing skin at defined time intervals. The distance between the nozzle end and the top of the tumor surface or normal skin was fixed at 18.5 mm (Figure 1A). Prior to sacrificing them, around 200–250 µL full blood was collected from the mandibular vein of the mice under anesthesia. The serological parameters were achieved using a FUJI dry-chem slide and dry-chem NX500 analyzer according to the protocols from the provider (Fujifilm, Tokyo, Japan). After sacrificing the animals, the tumor specimens and organs for histopathological evaluation were sampled and fixed in neutralized buffered formalin at 4 °C for 24 h. They were processed to generate paraffin-embedded sections for H&E staining and/or immunohistochemistry. The tumor tissues for molecular studies were frozen in liquid nitrogen until use. The animal study was approved by the Institutional Animal Care and Use Committee (IACUC) of National Yang Ming Chiao Tung University (IACUC approval no.: 1100422).

### 4.7. qPCR Analysis

The RNA from cells or tissues was isolated using TRI reagent (Molecular Research Center, Cincinnati, OH, USA). The expression of miRNA was measured using a TaqMan miRNA assay kit (Apply Biosystems, Waltham, MA, USA). The probes of the miRNAs, including miR-21, miR-31, miR-125b, miR-134, miR-146a, miR-187, and miR-211, together with the RNU6B (internal control) probe, are listed in Appendix A. The differential expression between the test samples and controls was calculated using the 2^−ΔΔCt^ method, where Ct is the cycle threshold of the fluorescence reaction [11].

### 4.8. Western Blot Analysis

An amount of 10 µg cell lysates isolated from cells or tumors was subjected to a Western blot analysis according to our published protocol [45]. The primary and secondary antibodies are described in Appendix A. The detection of the phosphorylation of Ser473 in AKT and the phosphorylation of Thr202/Tyr204 in ERKs delineated their activation states. The expression of the tested proteins was determined by normalizing to GAPDH.

### 4.9. Immunohistochemistry

The de-paraffined tumor tissue sections were subjected to immunohistochemical studies using the human specific anti-Ki-67 antibody (Agilent, Glostrup, Denmark) to detect the labeled tumor cells according to the protocol that we previously established [11].

### 4.10. Statistical Analysis

Data are presented as mean ± SE. The Mann–Whitney test and a two-way analysis of variance (ANOVA) were used for statistical analysis of the data. A *p* value less than 0.05 was considered significantly different.

## Figures and Tables

**Figure 1 ijms-24-16662-f001:**
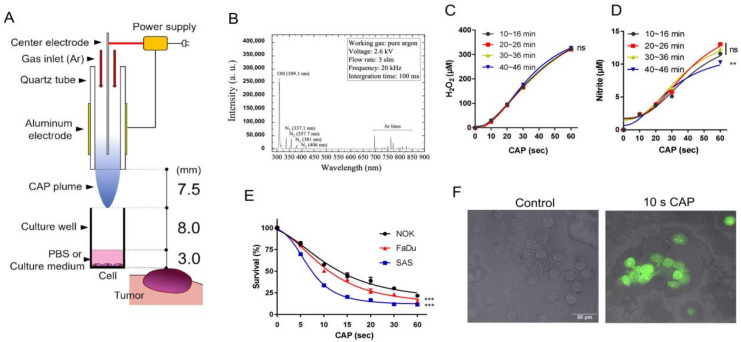
CAP irradiation induces RONS and cell death. (**A**). Schema to illustrate the CAP jet and the parameters. (**B**) OES spectrum of the CAP jet after 10 s CAP application on PBS. (**C**,**D**) H_2_O_2_ and nitrite concentration in PBS after 10 s, 20 s, 30 s, and 60 s CAP irradiation; the measurements were performed every 10 min and the CAP was switched on for 46 min during the measurement. (**E**) Survival of SAS, FaDu, and NOK cells 24 h after CAP irradiation for 5–60 s. (**F**) FaDu. Superimposition of bright-field microscopic images and fluorescence images. The induction of cellular ROS labeled by dichlorofluores *cin* diacetate (DCFDA) fluorescence in 10 s CAP irradiated cells relative to control. Bar, 50 µm. Data shown in (**C**–**E**) are mean ± SE of at least triplicate analysis. Two-way ANOVA test. ns, not significant; ** *p* < 0.01; *** *p* < 0.001. The data are the representatives of two individual assays.

**Figure 2 ijms-24-16662-f002:**
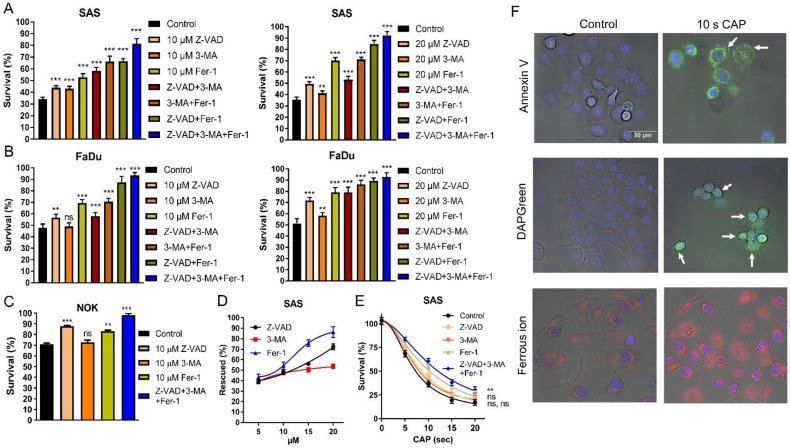
CAP irradiation induces apoptosis, autophagy, and ferroptosis. Pretreatment with ZVAD, 3-MA, Fer-1, and their combination as related to the cell survival following 10 s CAP irradiation. (**A**) SAS, (**B**) FaDu. Lt and Rt, 10 µM and 20 µM inhibitors, respectively. (**C**) NOK, 10 µM inhibitors. (**D**,**E**) SAS. (**D**) The cell survival being rescued by 5–20 µM inhibitors following 10 s CAP irradiation. (**E**) The cell survival following the 10 µM inhibition pretreatment and 5–20 s CAP irradiation. (**F**) FaDu. Superimposition of bright-field images and fluorescence images. Detection of Annexin V-labeled apoptotic cell (upper), the fluorescence-labeled autophagosome (middle), and ferrous ion (lower) induced by 10 s CAP irradiation. Nuclei are labeled with blue fluorescence. Bar, 50 µm. Arrows in the right upper panel indicate the presence of fluorescence on cell membranes. Arrows in the right middle panel indicate the presence of fluorescent vesicles in cells. Data shown in (**A**–**E**) are mean ± SE of at least triplicate analysis. Mann–Whitney test or two-way ANOVA test. ns, not significant; ** *p* < 0.01; *** *p* < 0.001. The data are the representatives of two individual assays.

**Figure 3 ijms-24-16662-f003:**
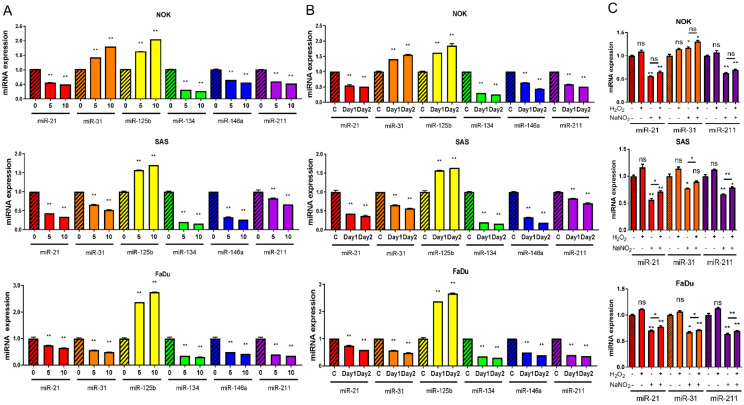
The expression of miRNAs in cells treated with CAP or chemicals. (**A**) Results of 5 s or 10 s CAP treatments, then growing cells for 24 h. (**B**) Results of 10 s CAP treatment, then growing cells for 24 h or 48 h. C, control. (**C**) Treatment with 12.5 µM H_2_O_2_ and/or 2 µM NO_2_^−^ for 1 h, then growing cell for 24 h. −, no treatment, +, treatment, Upper, NOK, Middle, SAS, Lower, FaDu. Data shown are mean ± SE of duplicate or triplicate analysis. Mann–Whitney test. ns, not significant; ** p* < 0.05; ** *p*< 0.01. The data are the representative of two individual assays.

**Figure 4 ijms-24-16662-f004:**
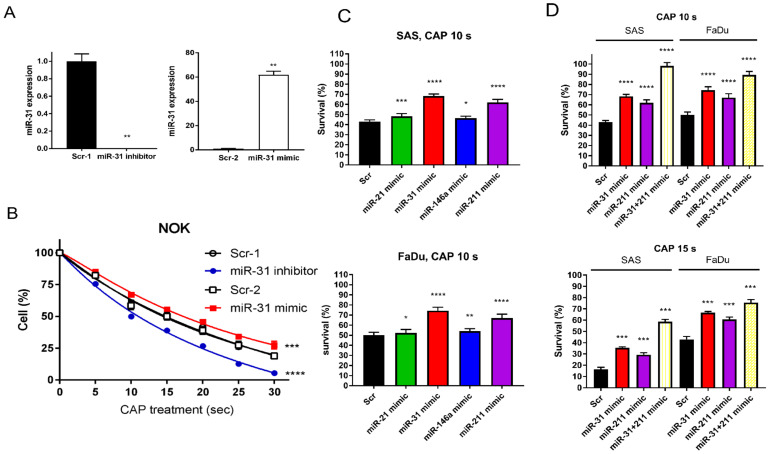
The CAP-induced OSCC cell death is rescued by miRNA mimics. (**A**,**B**) NOK treated with miR-31 inhibitor or mimic. (**A**) miR-31 expression analyzed by qPCR. (**B**) NOK survival following 5–30 s CAP irradiation, in the presence of miR-31 inhibitor or mimic. (**C**) The OSCC cell survives following 10 s CAP irradiation, in the presence of miR-21, miR-31, miR-146a, or miR-211 mimic. Upper, SAS; Lower, FaDu. (**D**) The OSCC cell survival following 10 s CAP irradiation (Upper) or 15 s CAP irradiation (Lower) in the presence of solitary miR-31 or miR-211 miRNA mimic, or their combination. Scr, scramble control. Data shown are mean ± SE of at least triplicate analysis. Mann–Whitney test or two-way ANOVA test. * *p* < 0.05; ** *p* < 0.01; *** *p* < 0.001; **** *p* < 0.0001. The data are the representatives of two individual assays.

**Figure 5 ijms-24-16662-f005:**
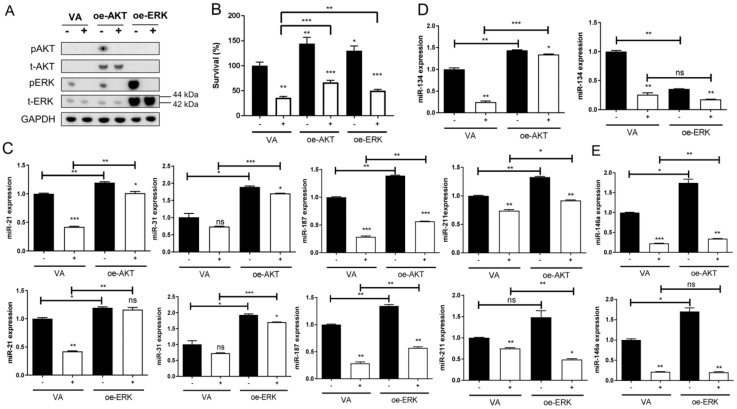
CAP irradiation inactivates AKT and ERK signals in SAS cells. SAS cells are transfected with plasmid in combination with CAP irradiation. (**A**) Western blot analysis. The endogenous and exogenous expression along with phosphorylation of AKT and ERK 2 h after CAP irradiation are analyzed. (**B**–**E**) Cell survival or miRNA expression 24 h later following signal activation and 10 s CAP irradiation. (**B**) Cell survival. (**C**) miR-21, miR-31, miR-187, and miR-211 expressions. (**D**) miR-134 expression. (**E**) miR-146a expression. oe-, transient overexpression; VA, vector alone control; p, phosphorylated; t-, total. −, control; +, CAP irradiation. Data shown are mean ± SE of duplicate or triplicate analysis. Mann–Whitney test. ns, not significant, * *p* < 0.05; ** *p* < 0.01; *** *p* < 0.001. The data are the representatives of two individual assays.

**Figure 6 ijms-24-16662-f006:**
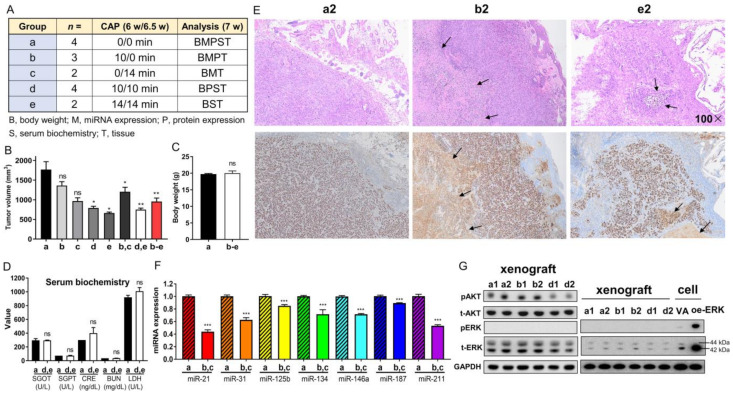
CAP irradiation inhibits the growth of SAS xenografts in nude mice. (**A**) Schema to illustrate the grouping, number of mice, CAP irradiation regimen, and the assays performed on the tumors and mice. a, control; b, c, one-shot CAP irradiation performed at 6 or 6.5 weeks; e, f, two-shot CAP irradiation performed at 6 and 6.5 weeks. (**B**) Tumor volume. (**C**) Body weight. (**D**) Serum biochemistry. (**E**) Representative hematoxylin and eosin (H&E)-stained histopathological diagrams of tumor tissue sections. Upper, H&E-stained tissue section. Lower, Ki-67 immunohistochemistry of the consecutive sections. Arrows indicate necrosis. Number, microscopic magnification fold. Scale bar: 100×. (**F**) The summary of miRNA expression from four control tumors and four tumors receiving one-shot CAP irradiation. Detailed data are integrated in Appendix A. (**G**) Western blot analysis on control tumors, and tumors receiving one-shot or two-shot CAP irradiation (two tumors in each group). Lt diagram encompasses AKT and ERK analysis. Note the absence of ERK activation in tumors. Rt diagram encompasses the concordant tissue and cell analysis within a blot to confirm the lack of ERK activation in tumors. The exposure time of the t-ERK panel for the Rt diagram is shorter than that for Lt diagram. oe-, transient overexpression; VA, vector alone control; p, phosphorylated; t-, total. Data shown are mean ± SE from at least triplicate analysis or from multiple samples in an experiment. Mann–Whitney test. ns, not significant, * *p* < 0.05; ** *p* < 0.01; *** *p* < 0.001.

## Data Availability

The data presented in this study are available on request from the corresponding author.

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
