# Peer review of "Cold Atmospheric Plasma Jet Irradiation Decreases the Survival and the Expression of Oncogenic miRNAs of Oral Carcinoma Cells"

_ijms, 2023, doi:10.3390/ijms242316662_

Round 1
Reviewer 1 Report
Comments and Suggestions for Authors
General Comments:
- The present manuscript is aimed at investigating the effects of Cold Atmospheric Plasma (CAP) treatment on oral squamous cell carcinoma (OSCC) cell lines, specifically: the impact of CAP on cell survival, miRNA expression, and signaling pathways is explored. Overall, the paper is well-structured, and the methods are adequately described. The experimental results appear to be well-supported by the data provided. The figures are well-constructed and add value to the paper. However, there are several areas where additional information and clarification are needed.
- Originality and Contribution: This study appears to be a significant contribution to the field, especially in understanding the effects of CAP on miRNA expression in OSCC cells. The combination of in vitro and in vivo experiments strengthens the findings.
Specific Comments:
- Abstract:
- Abbreviations should be mentioned in full the first time they appear.
- Introduction:
- The introduction provides a good overview of CAP and its potential applications in medical treatments. However, it needed to be more comprehensive and focus on cold plasma and ROS in relation to the proposed mechanisms of action.
- It would be beneficial to include a brief background on the current challenges and limitations associated with existing OSCC treatments. This would help to emphasize the significance of exploring alternative approaches such as CAP.
- Results:
- For ease of understanding, all figures should be made self-explanatory and presented in a simpler manner. This could be achieved by avoiding unnecessary repetition of results, as seen in Figure 4, where both (A), as well as (B, Rt) 10s CAP treatment, are mentioned.
- I would like to inquire about Figure 2, specifically regarding the 10s CAP treatment (24 h) depicted in the three graphs. Varying values for the expression of miR-31 in NOK across these graphs are observed. Additionally, it seems there may be unnecessary repetition of results. Could you kindly clarify this discrepancy?
- Clarify whether the data presented in the figures are representative of a single experiment or if it is a summary of multiple experiments.
- kindly consider adding statistical significance indicators to the figure captions, and not in the Materials and Methods section. Additionally, please add four asterisks’ explanations to the caption since they are present in some figures.
- In this reviewer’s opinion, the use of inhibitors alone is not sufficient to determine if CAP operates through the pathways of ferroptosis, apoptosis, or autophagy inhibition. It would be recommended to investigate specific markers' expression to support the results.
- Given the numerous studies that have established a connection between plasma's effect and the generated RONS, your findings indicating that CAP stimuli or products other than H2O2 may be responsible for triggering the disruption in miRNA expression are quite significant. I recommend conducting a repeat experiment, this time incorporating NO2 and a combination of H2O2 and NO2, to further explore this possibility.
- Discussion:
- I would like to inquire about the rationale behind employing SAS cells for evaluating CAP inactivation of AKT and ERK. It has been noted that SAS cells exhibit a moderate level of endogenous ERK activation and do not express AKT. Could you kindly provide further insight into this choice of cell line for the study?
- Materials and Methods:
o Please provide more detailed information on the CAP jet device. I observed in the referenced study in the results section some parameters. Nevertheless, upon examining Figure 1B, I noticed some additional parameters. To mitigate any potential confusion, it would be beneficial to incorporate this information within the context of the paper.
o kindly provide a description of the RONS measurements in CAP, including details about the fiber used, the range of detection, and the source of OES.
- Provide more details on the CAP treatment protocol, including the distance between the jet nozzle and the surface treated and any variability in treatment conditions.
- Please provide more detailed information on the pretreatment protocol with ZVAD, 3-MA, Fer-1, their combination, and miRNA inhibitors.
- Could you kindly offer additional details in the animal study section? Specifically, the CAP treatment protocol should be more detailed, including information on distance and timing. Furthermore, more information regarding sample collection and serologic tests.
Overall, the study presents interesting findings regarding the effects of CAP on OSCC cells. Addressing the above points will enhance the clarity and completeness of the manuscript.
Comments on the Quality of English Language- Language and Grammar:
- Carefully proofread the manuscript for language and grammar errors. There are some typos like, extra spaces in lines 73 and 74, line 84 containing one more letter at the beginning, subscript and superscript characters (writing the chemical formula of hydrogen peroxide H2O2 and formula for calculating tumor volume = 0.5ab2, the punctuation in different sections and figure captions
Author Response
Please see the attachment.
Q1. Abstract: Abbreviations should be mentioned in full the first time they appear.
A1. Thank you for your comment. We have accomplished the correction. Please refer to lines 21 – 24.
Q2. Introduction:
Q2-1) The introduction provides a good overview of CAP and its potential applications in medical treatments. However, it needed to be more comprehensive and focus on cold plasma and ROS in relation to the proposed mechanisms of action.
A2-1. Thank you. We have added more information about the CAP and ROS effects on cells. Please refer to lines 61 - 65 and the new references.
Q2-2) It would be beneficial to include a brief background on the current challenges and limitations associated with existing OSCC treatments. This would help to emphasize the significance of exploring alternative approaches such as CAP.
A2-2. Thank you for your critiques. We have improved the background description and incorporated additional information in Introduction section. Please refer to lines 86 - 98.
Q3. Results:
Q3-1) For ease of understanding, all figures should be made self-explanatory and presented in a simpler manner. This could be achieved by avoiding unnecessary repetition of results, as seen in Figure 4, where both (A), as well as (B, Rt) 10s CAP treatment, are mentioned.
A3-1. Thank you. We have corrected the repetition. The original Fig. 4A, which partially overlaps with Fig. 4B has been switched to supplementary Fig. S2 after revision.
Q3-2) I would like to inquire about Figure 2, specifically regarding the 10s CAP treatment (24 h) depicted in the three graphs. Varying values for the expression of miR-31 in NOK across these graphs are observed. Additionally, it seems there may be unnecessary repetition of results. Could you kindly clarify this discrepancy?
A3-2. Thank you for your questions. We apologize that A and B are partially overlapped in the original version. We have deleted the original A panel during the revision.
Q3-3) Clarify whether the data presented in the figures are representative of a single experiment or if it is a summary of multiple experiments.
A3-3. Thank you. All the repeated times in experiments and associated essential information have been described at the last part of each figure legend. Please refer to the revised manuscript.
Q3-4) kindly consider adding statistical significance indicators to the figure captions, and not in the Materials and Methods section. Additionally, please add four asterisks’ explanations to the caption since they are present in some figures.
A3-4. Thank you. We have removed the statistical significance indicators from the Materials and Methods section. Statistical significance indicators including four asterisks and the statistical test being used for each experiment have been integrated in associated figure legends.
Q3-5) In this reviewer’s opinion, the use of inhibitors alone is not sufficient to determine if CAP operates through the pathways of ferroptosis, apoptosis, or autophagy inhibition. It would be recommended to investigate specific markers' expression to support the results.
A3-5. We have performed additional experiments to show the induction of apoptosis, autophagy or ferroptosis using fluorescence-based marker detection. The methods are described in lines 443 - 450 and Table S1. The data are incorporated in revised Fig. 2F. The descriptions of the results are also incorporated in Results section lines 156 – 162 and associated figure legends.
Q3-6) Given the numerous studies that have established a connection between plasma's effect and the generated RONS, your findings indicating that CAP stimuli or products other than H2O2 may be responsible for triggering the disruption in miRNA expression are quite significant. I recommend conducting a repeat experiment, this time incorporating NO2 and a combination of H2O2 and NO2, to further explore this possibility.
A3-6. Thank you for your suggestions. To investigate the effects of nitrile in modulating miRNA expression, we treat cells with NaNO2. As viewed from Fig. 1D, about 2 µM NO2- is induced by 5s -10s CAP irradiation in PBS, cells are thus treated with H2O2 and/or NO2-for 1 h to assay the miRNA changes 24 h later. The analysis shown in revised Fig. 3C revealed that NO2– treatment causes the changes of miR-21, miR-31 and miR-211 expression similar to the effects of CAP irradiation. Please refer to Fig. 3C and its legend, the results in lines 189 – 193, and the brief discussion in lines 339 – 341.
Q4. Discussion: I would like to inquire about the rationale behind employing SAS cells for evaluating CAP inactivation of AKT and ERK. It has been noted that SAS cells exhibit a moderate level of endogenous ERK activation and do not express AKT. Could you kindly provide further insight into this choice of cell line for the study?
A4. Thank you for your comments. We have included your kind suggestions into the Discussion section to demonstrate the potential scope of this cell model. Please refer to lines 363 - 368.
Q5. Materials and Methods:
Q5-1) Please provide more detailed information on the CAP jet device. I observed in the referenced study in the results section some parameters. Nevertheless, upon examining Figure 1B, I noticed some additional parameters. To mitigate any potential confusion, it would be beneficial to incorporate this information within the context of the paper.
A5-1. Thanks for your comments. We have added the detailed information of CAP jet into text. Please refer to lines 101 – 107. We have also redrawn Fig 1A to illustrate better the theme of this CAP study.
Q5-2) kindly provide a description of the RONS measurements in CAP, including details about the fiber used, the range of detection, and the source of OES.
A5-2. Thanks for your comments. We have added the detailed information of OES measurement. Please refer to lines 107 – 115.
Q5-3) Provide more details on the CAP treatment protocol, including the distance between the jet nozzle and the surface treated and any variability in treatment conditions.
A5-3. Thank you. We have re-drawn Fig. 1A to illustrate more the essential CAP parameters including distance between nozzle head and fluid level, the cultivated cell and the tumor surface. In addition to Fig. 1A, the fundamental parameters associated with the functions of the appliance have been described in lines 101 – 115.
Q5-4) Please provide more detailed information on the pretreatment protocol with ZVAD, 3-MA, Fer-1, their combination, and miRNA inhibitors.
A5-4. We have incorporated more details of pre-treatment of cell death inhibitors and miRNA inhibitors. Please refer to lines 452 – 455 and 457 – 462, respectively.
Q5-5) Could you kindly offer additional details in the animal study section? Specifically, the CAP treatment protocol should be more detailed, including information on distance and timing. Furthermore, more information regarding sample collection and serologic tests.
A5-5. Thank you. We have incorporated the details of CAP, the collection of tumor samples and autopsy samples, and serologic methodology in animal studies in Material and Methods section. Please refer to lines 472 - 481.
Q6. Language and Grammar:
Carefully proofread the manuscript for language and grammar errors. There are some typos like, extra spaces in lines 73 and 74, line 84 containing one more letter at the beginning, subscript and superscript characters (writing the chemical formula of hydrogen peroxide H2O2 and formula for calculating tumor volume = 0.5ab2, the punctuation in different sections and figure cap
A6. Thank you for your comments. We have corrected the errors being pointed out by you during revision.

Reviewer 2 Report
Comments and Suggestions for Authors
In the MS ijms-2682901, the authors aim to attract the reader's interest by describing the anticancer effects of Cold Atmospheric Plasma (CAP) Jet on OSCC cell lines, deciphering the possible involved mechanisms. The authors analyzed CAP action through in vitro and in vivo studies. Their research is interesting and valuable, with potential application in OSCC therapy. The MS has 50 references, over half published in the last five years.
The Results Section is structured as follows:
2.1. CAP induces apoptosis, autophagy, and ferroptosis of OSCC cell lines
2.2. CAP downregulated the expression of oncogenic miRNAs in OSCC cells
2.3. The downregulation of oncogenic miRNAs underlies the CAP-induced decrease in cell survival
2.4. CAP inactivates AKT and ERK to reduce cell survival and impede miRNA expression
2.5. CAP treatment decreases the xenografic tumor growth of the SAS cell line
The Materials and Methods Section contains the following subsections:
4.1. Cell lines
4.2. Plasma jet device
4.3. Reagents
4.4. RONS measurement
4.5. qPCR analysis
4.6. Western blot analysis
4.7. Plasmids
4.8. Animal study
4.9. Immunohistochemistry
4.10. Statistical analysis
The following comments and suggestions are available below:
1. In the abstract, the authors offer a brief background (lines 16-19) and immediately jump to the general results of their study, presented in a general manner. The reviewer believes that a few phrases with the aim of the research and a short presentation of materials and methods are necessary. Moreover, the authors are encouraged to show a few concrete data from the results.
2. Introduction should follow the same structure as the abstract. The Introduction begins with the CAP description and anticancer effects in their current form. Then, the authors present the HNSCC and its frequent form OSCC, mixed with various cellular and molecular mechanisms, which play a role in tumor development. Next, they return to the CAP and its effects.
The reviewer encourages the authors to reorganize the data presented in the Introduction, showing first the general data about cancer, particularly with HNSCC and OSCC. Then, the current treatment and its limitations are followed by CAP potential.
Moreover, they should better indicate the previous literature research and the aim of the current one. They should better show the novelty of their study.
3. In Materials and Methods, the reviewer suggests restraining in a single subsection entitled "Materials" all data regarding CAT, all cell lines, laboratory animals, culture media equipment, and reagents. They are included in 3 subsections in the current form, and still more information is missing.
Moreover, some data from the Results are suitable for Materials and Methods (lines 84-88; 109-113; 116-119; 200-205). A similar comment is available for those from the Discussion regarding CAP structure (lines 250-252). More data regarding the cell lines are missing (provenance, type of OSCC, etc).
4. The authors measured RONS generated by CAP in PBS. Why did they not evaluate the oxidative stress in cells? Are they sure that the effects of CAP in PBS are the same in cells and that oxidative stress is the cause of apoptosis, ferroptosis, and autophagy?
5. The first subsection of Results is entitled "CAP induces apoptosis, autophagy and ferroptosis of OSCC cell lines" and shows the CAP structure, the RONS generated by CAP in PBS, the results of pretreatment of OSCC cells with inhibitors of apoptosis, autophagy, and ferroptosis, which underly the statement that CAP induces cancer cells death through these mechanisms.
6. The authors should add all data from results regarding the used procedure in materials and methods. The subsection 4.2. appears as results, not as Materials and Methods. The preliminary tests of the efficacy and stability of this system were performed on 1x phosphate-buffered solution (PBS, Biological Industries) and must be described (lines 345-347) with suitable references.
7. Suitable references must support the RONS measurement in PBS (lines 356-362).
8. They used 3 cell lines: 2 OSCC cells and 1 normal cells. However, the materials and methods used to obtain the results of subsection 2.1. are missing substantially. The same comment is available for cell death inhibitors. How did the authors purchase them?
9. Moreover, comparing the results obtained in tumor cells with normal ones is necessary, making the same tests on all 3 cell lines for comparison and uniformity of the study. Why only on OSCC cells?
10. Figure 2 is mentioned in the MS text in Subsection 2.2. - thus, the authors should place it after mention. In the current MS, Figure 2 is in Subsection 2.1., without any mention and relationship.
11. The authors must increase the lowest characters in Figures 1, 2, and 4 for better visualization.
12. The authors are invited to check and correct the entire MS to ensure all abbreviations are explained in the text at their first appearance.
Author Response
Please see the attachment.
Q1. In the abstract, the authors offer a brief background (lines 16-19) and immediately jump to the general results of their study, presented in a general manner. The reviewer believes that a few phrases with the aim of the research and a short presentation of materials and methods are necessary. Moreover, the authors are encouraged to show a few concrete data from the results.
A1. Thank you for your comments. We have largely re-organized and re-written the abstract to highlight the key rationales and findings of this study. Please refer to the revised abstract.
Q2. Introduction
Q2-1) Introduction should follow the same structure as the abstract. The Introduction begins with the CAP description and anticancer effects in their current form. Then, the authors present the HNSCC and its frequent form OSCC, mixed with various cellular and molecular mechanisms, which play a role in tumor development. Next, they return to the CAP and its effects.
Q2-2) The reviewer encourages the authors to reorganize the data presented in the Introduction, showing first the general data about cancer, particularly with HNSCC and OSCC. Then, the current treatment and its limitations are followed by CAP potential.
A2-1, 2. Thank you. The Introduction section has been largely re-organized to follow the structure being suggested by you.
Q2-3) Moreover, they should better indicate the previous literature research and the aim of the current one. They should better show the novelty of their study.
A2-3. Thank you. We have re-organized the Introduction section according to your opinions. In the last paragraphs in revised Introduction, we have described more clearly the crucial findings and values of this study. Please refer to lines 91 - 98.
Q3. In Materials and Methods:
Q3-1) the reviewer suggests restraining in a single subsection entitled "Materials" all data regarding CAT, all cell lines, laboratory animals, culture media equipment, and reagents. They are included in 3 subsections in the current form, and still more information is missing.
A3-1: Thank you. We have combined cell culture, reagents and plasmids into one subsection. However, since the animal study subsection has been expanded in response to reviewer #1’s opinions, we still retain this subsection as an independent one. Please refer to subsections 4.2 and 4.6 in the revised text.
Q3-2) Moreover, some data from the Results are suitable for Materials and Methods (lines 84-88; 109-113; 116-119; 200-205). A similar comment is available for those from the Discussion regarding CAP structure (lines 250-252). More data regarding the cell lines are missing (provenance, type of OSCC, etc.).
A3-2. Thank you, we have accomplished the correction according to your suggestions, and switched these sections being marked to Materials and Methods. Please refer to the appropriate parts in the text.
Q4. The authors measured RONS generated by CAP in PBS. Why did they not evaluate the oxidative stress in cells? Are they sure that the effects of CAP in PBS are the same in cells and that oxidative stress is the cause of apoptosis, ferroptosis, and autophagy?
A4. Thank you for your critiques. Based on the previous studies, oxidative stress is not the only factor that causes cell response. The other species like radicals or RNS also elaborate impacts on cells. However, it is difficult to measure all these species, so we only measured the H2O2 and nitrite as reference values in this study. Please refer to lines 111 – 115 and the related references 53 – 56 for details.
Q5. The first subsection of Results is entitled "CAP induces apoptosis, autophagy and ferroptosis of OSCC cell lines'' and shows the CAP structure, the RONS generated by CAP in PBS, the results of pretreatment of OSCC cells with inhibitors of apoptosis, autophagy, and ferroptosis, which underlie the statement that CAP induces cancer cells death through these mechanisms.
A5. Thank you. We have divided this subsection into two sections, i.e. new 2.1 and 2.2 after revision. The original Fig.1 has also been separated as new Fig.1 and Fig. 2 in the revised manuscript.
Q6-1. The authors should add all data from results regarding the used procedure in materials and methods. The subsection 4.2. appears as results, not as Materials and Methods.
A6-1. We have switched this section to Materials and Methods according to your comments. Thank you.
Q6-2. The preliminary tests of the efficacy and stability of this system were performed on 1x phosphate-buffered solution (PBS, Biological Industries) and must be described (lines 345-347) with suitable references.
A6-2. Thank you. We have described more in detail about the concerns raised by you. Please refer to lines 399 – 408 and reference paper 30. In addition, during revision we have performed assays to detect the cellular ROS using fluorescence-based approach. The cellular ROS induced by CAP-irradiation is shown in revised Fig. 1F and the associated contents in the text.
Q7. Suitable references must support the RONS measurement in PBS (lines 356-362).
A7. We have incorporated reference 30 to support the RONS measurement in PBS in the revised manuscript.
Q8. They used 3 cell lines: 2 OSCC cells and 1 normal cell. However, the materials and methods used to obtain the results of subsection 2.1 are missing substantially. The same comment is available for cell death inhibitors. How did the authors purchase them?
A8. The descriptions of death inhibitor treatment are re-organized in lines 452 - 455.
Q9. Moreover, comparing the results obtained in tumor cells with normal ones is necessary, making the same tests on all 3 cell lines for comparison and uniformity of the study. Why only on OSCC cells?
A9. Thank you. We have performed the inhibitor experiments in NOK. The data are integrated in the revised Fig. 2C, which indicates the rescue effects of V-ZAD and Fer-1. Please refer to the figure, legend the and the results incorporated in line 149 and 150.
Q10. Figure 2 is mentioned in the MS text in Subsection 2.2 - thus, the authors should place it after mention. In the current MS, Figure 2 is in Subsection 2.1, without any mention and relationship.
A10. We have corrected the wrong orders. Thank you.
Q11. The authors must increase the lowest characters in Figures 1, 2, and 4 for better visualization.
A11. Thank you. Fig. 1 has been separated into two figures to enrich the interpretation and resolution. The original Fig. 2A has been deleted. The remaining parts are amplified and the labels are also enlarged. Fig. 5A is switched to Fig. S2. The remaining panels and labels in this paper are amplified to improve the visualization.
Q12. The authors are invited to check and correct the entire MS to ensure all abbreviations are explained in the text at their first appearance.
A12. Thank you. We have corrected the concerns you raised.

Round 2
Reviewer 1 Report
Comments and Suggestions for Authors
All the comments have been addressed. According to this reviewer's opinion, the article can be published pending a minor revision of the English language.
Comments on the Quality of English LanguageAll the comments have been addressed. According to this reviewer's opinion, the article can be published pending a minor revision of the English language.
Author Response
Q1. All the comments have been addressed. According to this reviewer's opinion, the article can be published pending a minor revision of the English language.
A1. We have subjected this article to MDPI edition service (certificate no.: english-73811) for English edition. The language has been improved after edition. Thank you.

Reviewer 2 Report
Comments and Suggestions for Authors
The reviewer appreciates the authors' effort to revise the MS according to the Round 1 report, thus clarifying all presented data and improving the MS quality.
The reviewer considers that the authors suitably responded to all comments and suggestions.
Minor aspects remained to be corrected:
1. Please, separate the sub-subsection 2.3. title from the adjacent MS text (line 178).
2. Some parts of Figures 2, 3, 5, and 6 need to be increased because, in the current form, the characters are too small and do have not enough visibility.
Author Response
The reviewer considers that the authors suitably responded to all comments and suggestions. Minor aspects remained to be corrected:
Q1. Please, separate the sub-subsection 2.3. title from the adjacent MS text (line 178).
A1. Thank you. We have separated subsection 2.3 title into 2.3 and 2.4 after revision. Please refer to manuscript.
Q2. Some parts of Figures 2, 3, 5, and 6 need to be increased because, in the current form, the characters are too small and do have not enough visibility.
A2. We have amplified the diagrams and reduced the spaces between diagrams in Figures 2, 3, 5, and 6 to enrich visibility after revision. Please refer to the manuscript. Thank you for your comments.
